# *Downregulation of BUD31* Promotes Prostate Cancer Cell Proliferation and Migration via Activation of p-AKT and Vimentin In Vitro

**DOI:** 10.3390/ijms24076055

**Published:** 2023-03-23

**Authors:** Muhammad Choudhry, Yaser Gamallat, Ealia Khosh Kish, Sima Seyedi, Geoffrey Gotto, Sunita Ghosh, Tarek A. Bismar

**Affiliations:** 1Department of Pathology and Laboratory Medicine, Cumming School of Medicine, University of Calgary, Calgary, AB T2N 4N1, Canadayaser.gamallat@ucalgary.ca (Y.G.);; 2Departments of Oncology, Biochemistry and Molecular Biology, Cumming School of Medicine, University of Calgary, Calgary, AB T2N 4N1, Canada; 3Department of Surgery, Division of Urology, Cumming School of Medicine, University of Calgary, Calgary, AB T2N 4N1, Canada; 4Department of Medical Oncology, Faculty of Medicine and Dentistry, University of Alberta, Edmonton, AB T6G 2R7, Canada; 5Departments of Mathematical and Statistical Sciences, University of Alberta, Edmonton, AB T6G 2G1, Canada; 6Arnie Charbonneau Cancer Institute and Tom Baker Cancer Center, Calgary, AB T2N 4N1, Canada

**Keywords:** BUD31, Functional Spliceosome-Associated Protein 17, prostate cancer, ERG, p53, AR

## Abstract

Among men, prostate cancer (PCa) is the second most frequently diagnosed cancer subtype and has demonstrated a high degree of prevalence globally. BUD31, also known as Functional Spliceosome-Associated Protein 17, is a protein that works at the level of the spliceosome; it is functionally implicated in pre-mRNA splicing as well as processing, while also acting as a transcriptional regulator of androgen receptor (AR) target genes. Clinically, the expression of *BUD31* and its functions in the development and progression of PCa is yet to be elucidated. The BUD31 expression was assessed using IHC in a tissue microarray (TMA) constructed from a cohort of 284 patient samples. In addition, we analyzed the prostate adenocarcinoma (TCGAPRAD-) database. Finally, we used PCa cell lines to knockdown *BUD31* to study the underlying mechanisms in vitro.Assesment of BUD31 protein expression revealed lower expression in incidental and advanced PCa, and significantly lower expression was observed in patients diagnosed with castrate-resistant prostate cancer. Additionally, bioinformatic analysis and GSEA revealed that *BUD31* increased processes related to cancer cell migration and proliferation. In vitro results made evident that *BUD31* knockdown in PC3 cells led to an increase in the G2 cell population, indicating a more active and proliferative state. Additionally, an investigation of metastatic processes revealed that knockdown of *BUD31* significantly enhanced the ability of PC3 cells to migrate and invade. Our in vitro results showed *BUD31* knockdown promotes cell proliferation and migration of prostate cancer cells via activation of p-AKT and vimentin. These results support the clinical data, where low expression of BUD31 was correlated to more advanced stages of PCa.

## 1. Introduction

When looking globally, data from 174 countries revealed that 1,414,259 people have been diagnosed with prostate cancer (PCa), and this was reported alongside 375,304 prostate cancer related deaths [1]. Prostate cancer, globally, is the second most common form of cancer amongst men with an estimated incidence of 14.1%; this trails behind only to lung cancer which has an incidence rate of 14.3% [2]. Prostate cancer has been characterized as a heterogenous disease that can be further classified into distinct clinicopathological and molecular subtypes. PCa is a common malignancy in men, and its development and progression are complex processes that involve multiple genetic and environmental factors. Research has shown that the development of prostate cancer is associated with changes in the androgen signaling pathway [3], alterations in DNA repair mechanism [4], and dysregulation of various cellular signaling pathways [5]. The progression of prostate cancer from an androgen-dependent state to an androgen-independent state is a critical step in the development of advanced disease, and recent studies have shown that this transition is mediated by the activation of various growth factor signaling pathways, including the PI3K/Akt/mTOR pathway [3]. Additionally, research has identified several genetic and epigenetic alterations that contribute to the development and progression of prostate cancer, including alterations in the tumor suppressor genes PTEN and p53 [6], and the oncogene MYC [7]. The prostate gland, a component of the male reproductive system, is where prostate cancer develops. Prostate cancer can have a variety of effects on reproductive function, according to scientific research. Studies have found infertility can result from prostate cancer therapies such as surgery, radiation therapy, and hormone therapy because they can have a negative impact on sperm production and ejaculation [8]. PCa itself may also restrict the seminal vesicles, resulting in a reduction in the quantity of semen and sperm cells [9]. Additionally, prostate cancer may also result in erectile dysfunction, making it more difficult to initiate and maintain an erection, which may also result in infertility. Generally, the ability of men to reproduce and have children can be significantly impacted by prostate cancer [10]. Furthermore, as PCa is shown to be a highly prevalent disease worldwide, it is important to investigate and classify the roles of molecular markers in its development and progression, with the aim of revolutionizing current methods for clinical diagnosis and treatment. 

Recent studies have revealed that mutations in spliceosome components are present in various types of cancer, including breast, lung, and leukemia [11]. Mutations in SF3B1, SRSF2, and U2AF1, for example, have been identified in myelodysplastic syndromes, chronic lymphocytic leukemia, and other hematological malignancies [12]. These mutations often result in altered splicing patterns that promote tumor growth and survival. One of the ways that the spliceosome promotes cancer development is through the production of oncogenic isoforms [13]. Alternative splicing can lead to the generation of protein isoforms that promote tumor growth and metastasis. For example, the spliceosome machinery can promote the inclusion of exons that encode for protein domains that enhance cell proliferation, invasion, and migration. In some cases, aberrant splicing can also lead to the production of non-functional or truncated proteins that promote cancer cell survival and resistance to treatment [12]. Furthermore, spliceosome alterations can also affect the expression of genes involved in DNA damage response and repair [11]. This is particularly important in the context of cancer therapy, where the DNA damage response is often targeted to induce apoptosis in cancer cells. Spliceosome mutations can alter the splicing of genes involved in DNA repair, leading to increased resistance to chemotherapy and radiation therapy. In addition, spliceosome mutations can also affect the splicing of genes involved in the cell cycle, leading to the deregulation of cell proliferation and survival [14].

Interestingly, recent studies have also shown that the spliceosome machinery can be targeted as a therapeutic strategy for cancer treatment [11]. Spliceosome inhibitors, such as spliceostatin A and H3B-8800, have been shown to induce apoptosis in cancer cells with spliceosome mutations, suggesting a potential therapeutic benefit for patients with spliceosome-related cancers. However, as with all targeted therapies, it is important to consider potential off-target effects and the development of resistance mechanisms.

BUD31 is a protein coding gene that was determined to exhibit subcellular localization in the nucleus; further studies on this topic were able to uncover that BUD31 was more concentrated in the promoter regions of androgen receptor (AR) target genes on chromatin [15,16]. Currently, *BUD31* has been shown to be involved in processes including transcriptional regulation and mRNA processing by functioning as part of the spliceosome as a splicing factor [17,18]. The spliceosome is a ribonucleoprotein complex that is responsible for carrying out splicing of pre-mRNA [19]. Using yeast as a model organism, studies have determined that Bud31 associates with 3 small nuclear ribonucleoprotein (snRNP) complexes that are involved in RNA splicing, the interacting snRNPs are referred to as U2, U5, and U6 spliceosomal complexes [20,21]. Additionally, it was also shown that Bud31 in yeast models facilitated cell cycle progression through the G1/S checkpoint [22]. In recent years, researchers have uncovered the emerging role of the spliceosome in the development and progression of many cancer subtypes, where cancer-specific splicing events and dysregulation of splicing factor genes have been shown to be involved in disease progression [23]. Moreover, another study found that *BUD31* and the spliceosome are necessary for progression and metastasis in MYC-driven breast cancer cells [24]. Interestingly, BUD31 has been implicated as a co-regulator of AR transcriptional activity in prostate cancer cell lines [16]. 

In this study, we investigate the clinical implications of varying outcomes relative to *BUD31* expression levels by using a cohort of tissue microarrays (TMAs) and data from The Cancer Genome Atlas (PRAD-TCGA) to analyze how expression of this gene varies in patients with different tumor stages and mutations. Then, we further elucidated the function of *BUD31* in PCa by inducing reduced expression of the gene through in vitro methods utilizing RNA interference. 

## 2. Results

### 2.1. BUD31 Expression in Clinical Cohort with Prostate Cancer

Within our study, we aimed to investigate the expression of BUD31 protein in different stages of prostate cancer, including benign, incidental PCa, advanced PCa, and castrate-resistant PCa (CRPCa). To achieve this, we performed an immunohistochemistry (IHC) analysis on our own cohort, which consisted of 284 patients (as shown in Figure 1A). Our analysis revealed that there was a significant decrease in BUD31 expression in CRPCa compared to the benign stage (as illustrated in Figure 2B), with a *p*-value of 0.018. This finding indicates that BUD31 may play a crucial role in the development and progression of prostate cancer and that a decrease in its expression may be associated with the more aggressive and advanced stages of the disease. Overall, these results provide valuable insights into the potential diagnostic and prognostic significance of BUD31 in prostate cancer and highlight the importance of further investigation into its role in the disease.

### 2.2. Low Expression of BUD31 Is Associated with Worse Overall Survival and Return Free Survival

In our study, our initial focus was to examine whether low BUD31 expression was prevalent in other types of cancers. Upon analyzing patients diagnosed with prostate adenocarcinoma, we found no significant difference in overall survival between groups with high and low BUD31 expression (as depicted in Figure 2A). However, we observed that clinical data from patients diagnosed with testicular germ cell tumor, ovarian cancer, and lung squamous cell carcinoma indicated that low BUD31 expression was strongly associated with worse overall survival (OS) (as illustrated in Figure 2B–D). Furthermore, we also found that low BUD31 expression was related to worse return-free survival (RFS) in patients diagnosed with kidney renal papillary cell and thyroid carcinoma (as demonstrated in Figure 2E,F). Therefore, our investigation suggests that low BUD31 expression is a potential prognostic factor for predicting overall survival and RFS as shown in several other types of cancers, including testicular germ cell tumor, ovarian cancer, lung squamous cell carcinoma, kidney renal papillary cell carcinoma, and thyroid carcinoma.

### 2.3. BUD31 Gene Set Enrichment Analysis in TCGA PRAD Database

Next in our study, we conducted a gene set enrichment analysis (GSEA) on BUD31 in a TCGA PRAD cohort to gain a deeper understanding of its associated gene set. As shown in Figure 3A, the GSEA data for BUD31 depicts the profiled gene set associated with BUD31. Additionally, we generated two heat maps to illustrate the positively and negatively correlated genes as well as listing the 50 most significant genes in each subset (Figure 3B,C). Our analysis of the GSEA revealed the downregulation of pathways related to several biological processes, including cell junction organization, cell-substrate adhesion, maintenance of cell number, cell morphogenesis, and TGF-beta production (as depicted in Figure 3D). We also found that genes and pathways pertaining to the basal component of the cell were the only cellular processes that were significantly downregulated (as illustrated in Figure 3E), while molecular processes such as p53 and protein kinase A binding were significantly downregulated (as shown in Figure 3F). To provide a broader and more integrated approach to the GSEA, we categorized the most enriched processes in biological, cellular, and molecular contexts and visually summarized them on bar plots (as illustrated in Figure 3E). These results suggest that BUD31 may play a crucial role in regulating several key biological, cellular, and molecular processes in prostate cancer, providing valuable insights into its potential diagnostic and therapeutic significance for the disease. Further investigation is needed to elucidate the exact mechanisms underlying these findings and their potential clinical implications.

### 2.4. BUD31 Regulates Cell Cycle Progression of Prostate Cancer Cells In Vitro

The knockdown efficiency of BUD31 in PC3 and LNCaP cells was further analyzed to better understand the underlying mechanisms involved in the dysregulation of cell cycle progression and apoptosis. To confirm the knockdown efficiency of BUD31, Western blot analysis was performed, and a significant decrease in the expression of BUD31 protein was observed in both PC3 and LNCaP cells (Figure 4A). Quantification of these results showed a 70% decrease in the expression of BUD31 protein in PC3 cells, and an 80% reduction in LNCaP cells, indicating a robust and efficient knockdown of BUD31 in both cell lines (Figure 4A). To investigate the role of BUD31 as a cell cycle regulator, various cell cycle markers were examined using flow cytometry. The results showed that 48 h after BUD31 was knocked down, there was a 4.66% decrease in the number of cells in G1, while there was a 1.91% increase in the number of cells in G2 when compared to the negative control treatment. Conversely, there was also a 2.76% increase in the number of cells found in S phase when comparing the knockdown to the control groups (Figure 4B). These results suggest that BUD31 downregulation leads to dysregulated cell cycle progression, with cells progressing more rapidly through the G1 and S phases, and accumulating in the G2 phase.

Furthermore, the effect of BUD31 downregulation on apoptosis was also examined. It was observed that cells with BUD31 downregulation demonstrated a 1.8% increase in apoptotic cell populations relative to the negative control treatment (Figure 4C). These findings suggest that BUD31 may also play a role in regulating apoptosis. The dysregulation of cell cycle progression and apoptosis led us to explore the expression of multiple cell cycle regulator proteins. AKT signaling activity was analyzed using Western blot, which showed that the downregulation of BUD31 resulted in significant upregulation of p-AKT in PC3 cells but not LNCaP cells (Figure 4D). This increase was shown to be quite tremendous in siBUD31 treated PC3 cells as it was calculated to have almost 3-times the expression of P-AKT when compared to the control group (Figure 4D). However, in LNCaP cells, when BUD31 was knocked down, it resulted in a significant reduction in p53 protein, where expression was reduced by approximately 40% (Figure 4D). These results suggest that BUD31 may regulate the expression of multiple cell cycle regulator proteins in a cell type-specific manner.

### 2.5. Inhibition of BUD31 Enhances the Migration and Invasion of PCa 

Metastasis is a key hallmark of cancer progression and understanding the role of BUD31 in this process is of great importance. To further investigate the effects of BUD31 inhibition on the metastatic phenotype of PCa cells, additional experiments were conducted. To elucidate whether BUD31 plays a role in producing a metastatic phenotype, migration and invasion assays were performed to determine whether the inhibition of BUD31 would amplify or inhibit the PCa cells’ ability to migrate and invade. First, when looking at the results of the migration assay it was shown that the downregulation of BUD31 led to the enhanced migration of PC3 cells, but not for the PC3-ERG cell line (Figure 5A). In PC3 cells, this difference was quantified and graphed to demonstrate the 2.5-fold increase in migratory cells when treated with siBUD31 (Figure 5B). Alternatively, in PC3-ERG cells, siBUD31-treated cells showed no significant difference in the number of migratory cells relative to the control treatment (Figure 5B). Similar results were found when exploring the effects of BUD31 knockdown on the invasion of cells, where PC3 cells experiencing the downregulation of BUD31 exhibited increased invasion, and the fold change was quantified to be approximately two times that of the PC3 control cells (Figure 5C,D). However, PC3-ERG cells demonstrated no significant difference in their invasive ability when subjected to loss of BUD31 (Figure 5C,D). When analyzing common markers for driving epithelial-mesenchymal transition (EMT), it was observed that in PC3 cells vimentin demonstrated a 75% increase in expression in the siBUD31-treated cells (*p* < 0.0001), whereas N-cadherin levels remained unchanged (Figure 5E,F). 

## 3. Discussion

In the current study, we observed that low BUD31 protein expression was associated with more severe PCa subtypes, and the lowest expression was found in patients diagnosed with castrate-resistant PCa. From the TCGA-PRAD database, we observed no significant difference in the overall survival between groups with high or low BUD31 expression. However, we were able to correlate low BUD31 expression to worse overall and return-free survival in multiple other cancer subtypes. 

Gene set enrichment analysis of *BUD31* revealed the downregulation of pathways involved in regulating cell migration and cell cycle progression, these processes included cell adhesion and p53 binding. These results suggest that loss of BUD31 would impact the cell cycle and cytoskeletal dynamics, producing a more proliferative and invasive cancer cell [25]. The results of the GSEA may indicate a specific genomic signature found in cancer cells of more advanced PCa stages. Currently, no direct studies have been published regarding any potential relationship that might exist between *p53* and *BUD31*. This led us to execute in vitro studies to further investigate the role of *BUD31* with *p53*. 

Our in vitro studies revealed that BUD31 downregulation is associated with aggressive tumor cell behavior when compared to wildtype expression in PC3, PC3-ERG, and LNCaP cell lines. Flow cytometry results revealed that BUD31-depleted PC3 cells exhibited an increase in the G2 cell population, which indicates a greater number of cells undergoing mitotic divisions, and therefore is evidence of a more proliferative state when cells are treated with si*BUD31*. Furthermore, knockdown of *BUD31* revealed varying rates of apoptosis where cells treated with siBUD31 had a 1.8% increase in cell death. Looking at common cell cycle markers and signaling pathways, we observed that when BUD31 expression was decreased in PC3 cells, the AKT pathway was hyperactivated, showing greater levels of its active-phosphorylated state. This strengthened AKT signaling is thought to aid cell cycle progression, similarly to how it was reported in LNCaP cells and Hodgkin’s lymphoma-derived cell line [26]. In addition, we noticed a significant decrease in p53 expression observed in the siBUD31-treated LNCaP cell line. This decreased expression of p53 supports our data, where there was a decrease in the population of cells arrested in G1; this is in agreement with parallel studies that also find p53 expression correlated with G1 cell cycle arrest [27]. In retinoblastoma, scientists found that the loss of p53 did result in increased G2 cell populations [28]. 

When looking at the results from migration and invasion, it was evident that loss of BUD31 led to the enhanced migratory and invasive nature of PC3 cells. Focusing on the PC3 cell line, the expression of EMT markers were analyzed which revealed a significant increase in the expression of vimentin. In tumor cells, researchers identified the overexpression vimentin as a major driver of epithelial-mesenchymal transition process, leading to poorer clinical outcomes [29]. This function was also found in PCa as studies in the field found that the upregulation of vimentin led to increased cancer cell invasiveness; this trend was also observed in our study’s in vitro experiments when BUD31 was knocked down [30]. 

Furthermore, a limitation of this study involves the lack of heterogeneity found in the in vitro studies and, hence, why future studies may be necessary to elaborate on the nature of BUD31 expression in different molecular signatures across a variety of different tumor cell types. 

## 4. Material and Methods

### 4.1. Study Population and Pathological Analysis

Tissue micro-arrays (TMA) were constructed from a cohort of 284 patients consisting of benign, incidental, advanced and castrate-resistant PCa. BUD31 expression was assessed in relation to Gleason grade. The diagnosis of individual cores from the TMA was confirmed by histological analysis performed by the study pathologist (TAB). BUD31 expression levels were classified using a three-tier system (0, negative; 1, weak; 2, moderate; and 3, high intensity). Gleason grades were assessed according to the 2018 WHO and ISUP grade group by the study pathologist (TAB).

### 4.2. Immunohistochemistry (IHC)

The protein BUD31 expression was assessed through IHC staining. The Dako Omnis Auto Stainer was used to stain 4 µm formalin-fixed paraffin-embedded (FFPE) sections using the standard protocol (DAKO Omnis Stainer User Manual (2019) (Agilent Technologies, Santa Clara, CA, USA). Tissue sections were cut and mounted on microscope slides, which were deparaffinized and rehydrated using gradually decreasing concentrations of ethanol. Antigen retrieval was performed using citrate epitope retrieval buffer (pH 6.0). Rabbit monoclonal BUD31 antibody (1:50) Cat# Sc-374163 (Santa Cruz Biotechnology, Santa Cruz, CA, USA) was used. The detection reagent used was the FLEX DAB+ Substrate Chromogen system Cat# GV82511-2 (Agilent Technologies, Santa Clara, CA, USA).

### 4.3. TCGA PRAD Data Analysis 

To investigate the expression of *BUD31* from the prostate adenocarcinoma (TCGA PRAD) database found in The Cancer Genome Atlas (TCGA) program, this cohort comprised 497 male patients (n = 497). The bioinformatic analysis utilized LinkedOmics to analyze the RNAseq database extracted from TCGA [31]. This online tool utilizes rapid analysis servers to compare RNAseq data for a specific gene of interest. Furthermore, we analyzed RNA expression of the *BUD31* in normal and tumor tissue from the TCGA PRAD database for prostate cancer-specific expression. 

To investigate the biological, cellular, and molecular consequences of abnormal *BUD31* expression, we utilized LinkedOmics (http://www.linkedomics.org (accessed on 17 October 2022) to generate a Gene Set Enrichment Analysis (GSEA) from the TCGA PRAD database. This analysis was based on FDR and used a web-based toolkit and explorer.

### 4.4. Cell Lines

The human prostate cancer cells line used in this study include PC3, PC3-ERG, LNCaP, and DU145 cells. All cell lines were purchased from the American Type Culture Collection (ATCC; Manassas, CA, USA). Stable PC3-ERG cell lines were obtained from Felix Feng, University of Michigan [32].

The PC3 and PC3-ERG prostate cancer cells were cultured in DMEM/F12 (GIBCO life technology, Grand Island, NY, USA) enriched with 10% FBS (GIBCO life technology, Grand Island, NY, USA) and 1% Pen Strp (penicillin/streptomycin) (Ref # 15140-122, GIBCO life technology, Grand Island, NY, USA).

The LNCaP prostate cancer cell lines were cultured in RPMI-1640 medium (GIBCO life technology, Grand Island, NY, USA). DU145 cell lines were grown in DMEM media (GIBCO life technology, Grand Island, NY, USA). Additionally, all the media stated previously was supplemented with 10% FBS (GIBCO life technology, Grand Island, NY, USA) and 1% Pen Strp (penicillin/streptomycin (Ref# 15140-122, GIBCO life technology, Grand Island, NY, USA). The cells were incubated at 37 °C in a 5% CO_2_ atmosphere.

### 4.5. Cell Line Transfection and RNA Silencing 

The knockdown of *BUD31* was performed using small interfering RNAs. A pre-designed siRNA silencer and a scrambled siRNA (as negative control) were obtained from Ambion, Grand Island, NY, USA. Briefly, PC3 and LNCaP cells were plated in six well plates until they reached about 70–80% confluency. Next, the reaction mix for the transfection of siRNA was prepared using Opti-MEM (GIBCO life technology, Grand Island, NY, USA) and Lipofectamine RNAiMAX (Invitrogen, Carlsbad, CA, USA) according to the manufacturer’s instructions. A Western blot analysis was performed to confirm the knockdown of BUD31, while also assessing the efficiency and duration of the transient BUD31 knockdown. 

### 4.6. Western Blot

Total protein was extracted using RIPA lysis buffer (Cat #9806, Cell Signaling, Danvers, MA, USA) with the addition of 1:100 protease PMSF inhibitors (Cat # 5872S, Cell Signaling, Danvers, MA, USA). When running the Western blot, equal quantities of proteins were loaded into each well and separated by size on a polyacrylamide SDS gel. Then, the proteins were transferred to a PVDF membrane (BIO-RAD Immuno-Blot^®^ Membrane). Once the transfer was complete, nonspecific binding was blocked by incubating the membrane on a shaker with a blocking buffer comprised of 10% skimmed milk in TBS for 1 h at room temperature. The membranes were then incubated with a primary antibody (Appendix A) at 4 °C overnight. This was followed by incubation with either anti-rabbit IgG or anti-mouse IgG secondary antibody conjugated to HRP horseradish peroxidase (Cell Signaling, Danvers, MA, USA) in antibody dilution buffer for 1 h at 37 degrees Celsius. Washing steps were performed after primary and secondary antibody incubations, which involved 3 washes with TBS buffer + 1% Tween for 5 min each. The ECL substrate Chemiluminescence signal was detected using the ChemiDoc imaging system (Bio-Rad Laboratories, Hercules, CA, USA).

### 4.7. Migration and Invasion Assay

The PC3 cell lines were knocked down as previously described. Twenty-four hours after transfection the cells were trypsinated, then transferred to either the top of a Corning Biocoat control inserts for the migration assay (Ref # 354578, Corning, Bedford, MA, USA), or a Corning Matrigel invasion chamber (Ref# 354480, Corning, Bedford, MA, USA) for the invasion assay. After 48 h elapsed, the cells were fixed and stained with Diff Quick (Siemens Healthcare diagnostics, Tarrytown, NY, USA). Images were then developed by using the inverted EVOS FL Life microscope, images were captured in the bright field with 10× and 40× magnifications. The number of cells were quantified by using multiple frames which are counted, then averaged from 40× magnification and compared to the negative control for analysis. 

### 4.8. Flow Cytometry 

For investigating the role of BUD31 on cell cycle and apoptosis, we performed flowcytometry cell cycle and apoptosis analysis. Briefly, *BUD31* knockdown and the corresponding control groups with the appropriate number of replications were prepared as previously described for the knockdown. They were further harvested, washed in cold PBS, fixed in 70% ethanol, and stained with FxCycle™ PI/RNase Staining Solution containing 50 µg/mL propidium iodide and 100 µg/mL RNase A in PBS (Cat # F10797, Invitrogen, Carlsbad, CA, USA). The cells were analyzed for their DNA content with a BD LSR II Flow Cytometer. For the Annexin V/PI assay apoptosis assay, cells were prepared as previously stated. Cells were then treated with Alexa Fluor 488 Annexin V/Dead Cell Apoptosis Kits as per manufacturer instruction (Cat # V13241, Invitrogen, Carlsbad, CA, USA). The results were analyzed using BD LSR II Flow Cytometer. The data were further analyzed using FlowJo™ v10 Software-BD Biosciences.

### 4.9. Statistical Analysis

Statistical analysis was performed using GraphPad Prizm (v 9.1.0). The unpaired t-test was applied to compare between the two groups. All values were provided as Mean ± SEM or Mean ± SD. *p* value < 0.05 were considered significant. 

## 5. Conclusions

To conclude, our study acknowledges the role of BUD31 as a tumor suppressor as depicted by the results of our clinical and in vitro studies. Clinical data negatively correlated BUD31 expression with overall and return-free survival. Our in vitro results supported these findings as the knockdown of BUD31 increased prostate cancer cell proliferation and migration. The activation of AKT and reduction of p53 expression was implicated in the increased proliferative state of prostate cancer cells. Additionally, increased vimentin expression was found to enhance the migratory and invasive nature of PCa cells in vitro.

## Figures and Tables

**Figure 1 ijms-24-06055-f001:**
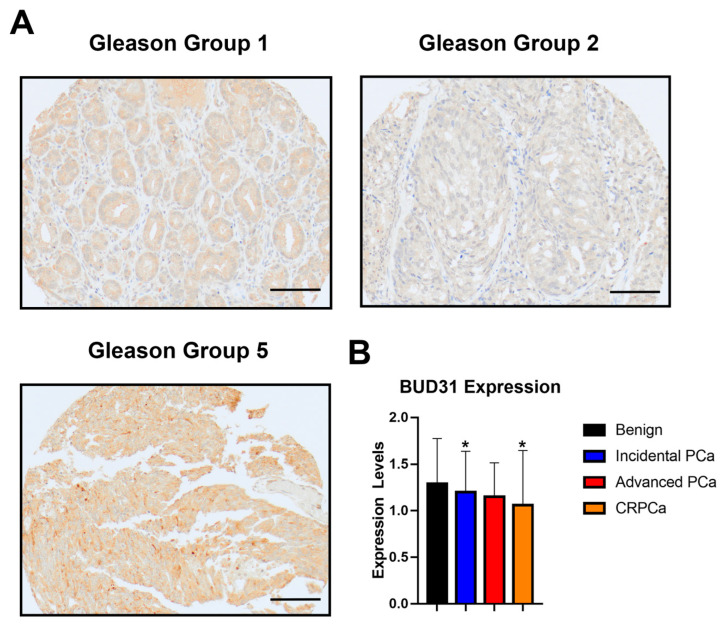
Expression of BUD31 in clinical cohort of patients diagnosed with prostate cancer. (**A**) Immunohistochemistry staining images of BUD31 expression in Gleason grouping, images were observed under 20× magnification (scale bar = 100 μm). (**B**) Bar plots show the BUD31 expression in benign (n = 76), incidental (n = 138), advanced (n = 20) and CRPC (n = 50) clinicopathological subtypes of prostate cancer. The data’s significance was determined by using the students *t*-test and it is plotted along with its Mean ± SD bars, (*) denotes a *p* value less than 0.05.

**Figure 2 ijms-24-06055-f002:**
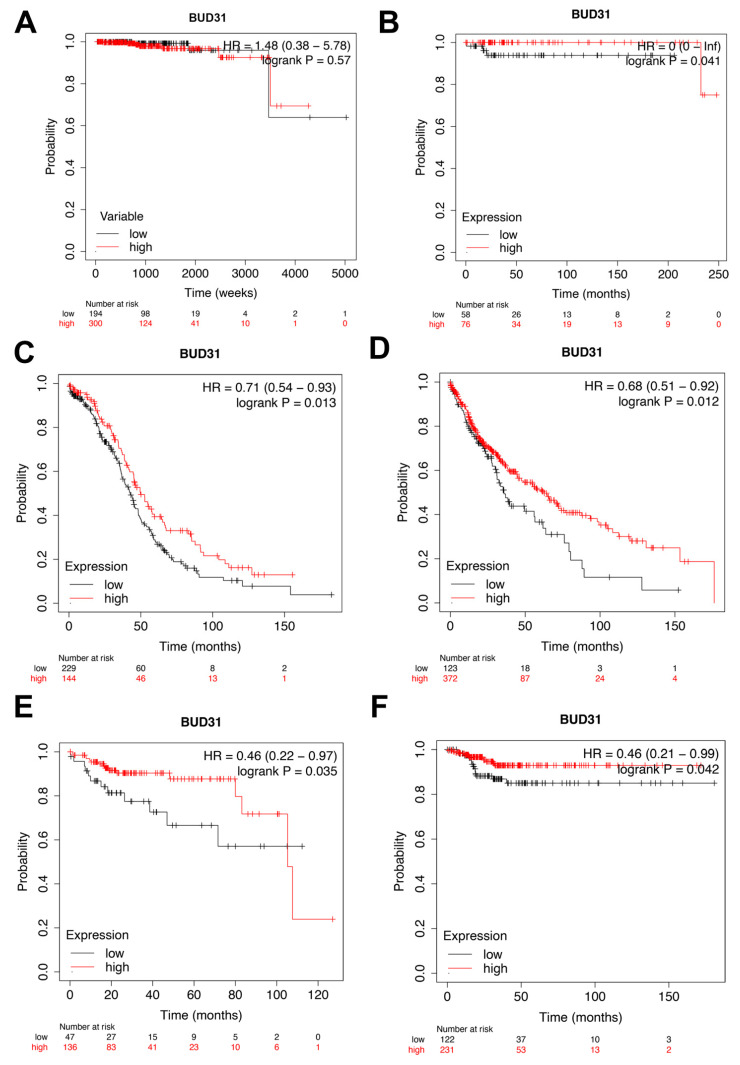
Kaplan–Meier plots indicating overall and return-free survival in relation to BUD31 expression in different cancers. (**A**) Prostate adenocarcinoma (OS). (**B**) Testicular germ cell tumor (OS). (**C**) Ovarian cancer (OS). (**D**) Lung squamous cell carcinoma (OS). (**E**) Kidney renal papillary cell carcinoma (RFS). (**F**) Thyroid carcinoma (RFS).

**Figure 3 ijms-24-06055-f003:**
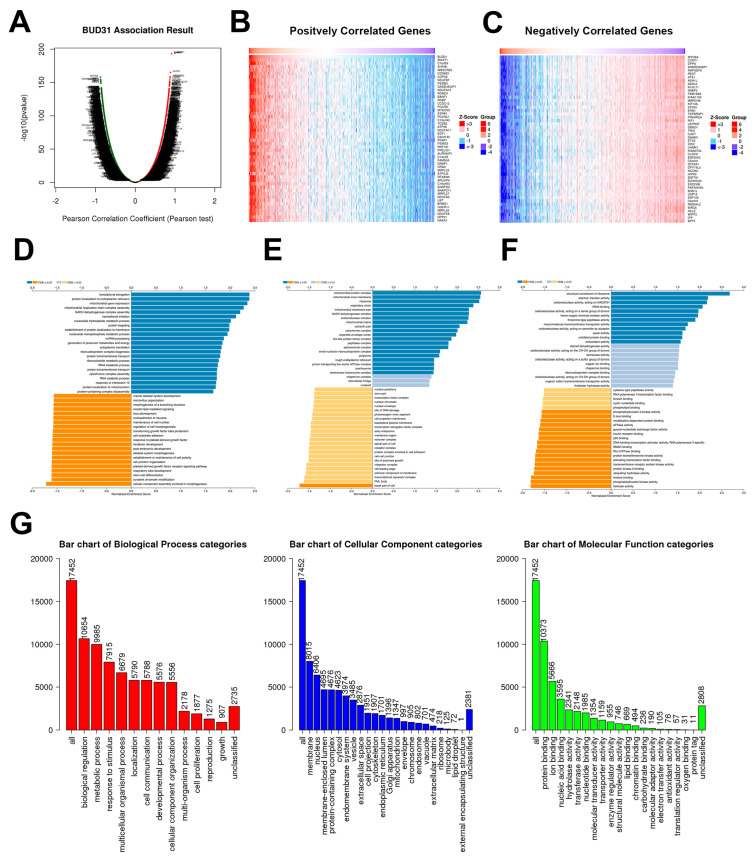
Gene set enrichment analysis for *BUD31* from TCGA-PRAD RNA-seq of prostate cancer patients. (**A**) Volcano plot showing *BUD31*-correlated genes, where red dots depict upregulated genes with positive Pearson correlation coefficients, whereas green dots show downregulated genes with negative Pearson correlation coefficients (FDR < 0.01). (**B**,**C**) Heatmap analysis demonstrating the most upregulated and downregulated genes. (**D**–**F**) Gene ontology (GO) enrichment analysis depicting the most positively and negatively correlated biological, cellular, and molecular processes, respectively. Ranking was done using Pearson correlation coefficients and *FDR* values. The *FDR* is calculated using the Benjamini-Hochberg method. (**G**) GSEA summary of the most upregulated biological, cellular, and molecular processes, ranked using *FDR* and number of IDs in the cohort.

**Figure 4 ijms-24-06055-f004:**
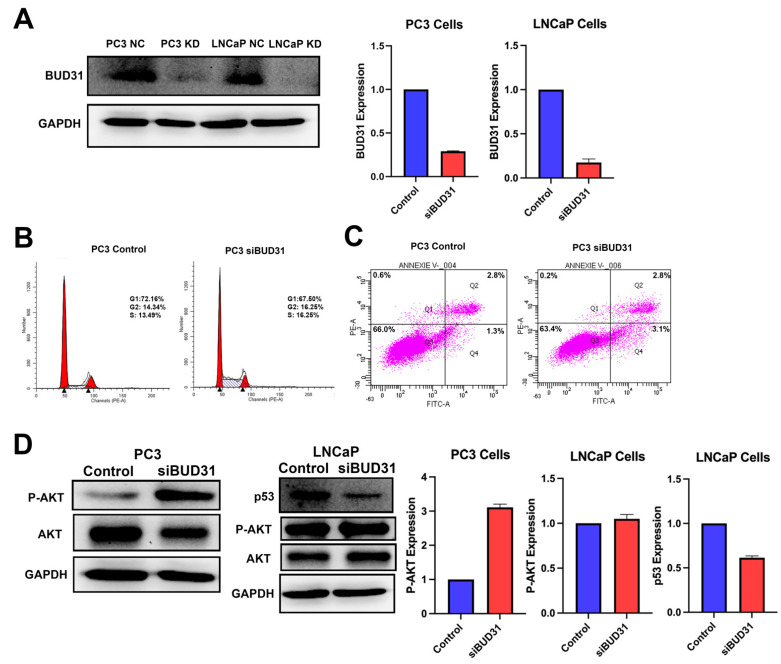
Knockdown of *BUD31* leads to dysregulation of cell cycle progression and signaling. (**A**) Western blot analysis and bar plot showing decreased BUD31 expression in siBUD31 treated PC3 and LNCaP cell lines. (**B**) Cell cycle analysis from 3 replicates of flow cytometry for PC3 control and si*BUD31* treated cells using FxCycle PI/RNase Staining Solution. (**C**) Flow cytometry assessment of apoptotic cells in control and si*BUD31* PC3 cells from 3 replicates, analysis was done using Alexa Fluor^®^ 488 annexin V/Dead Cell Apoptosis Kit with Alexa^®^ Fluor 488 annexin V and PI. (**D**) Western blot analysis and graphical representation of p-AKT and p53 expression for PC3 and LNCaP cell lines, respectively.

**Figure 5 ijms-24-06055-f005:**
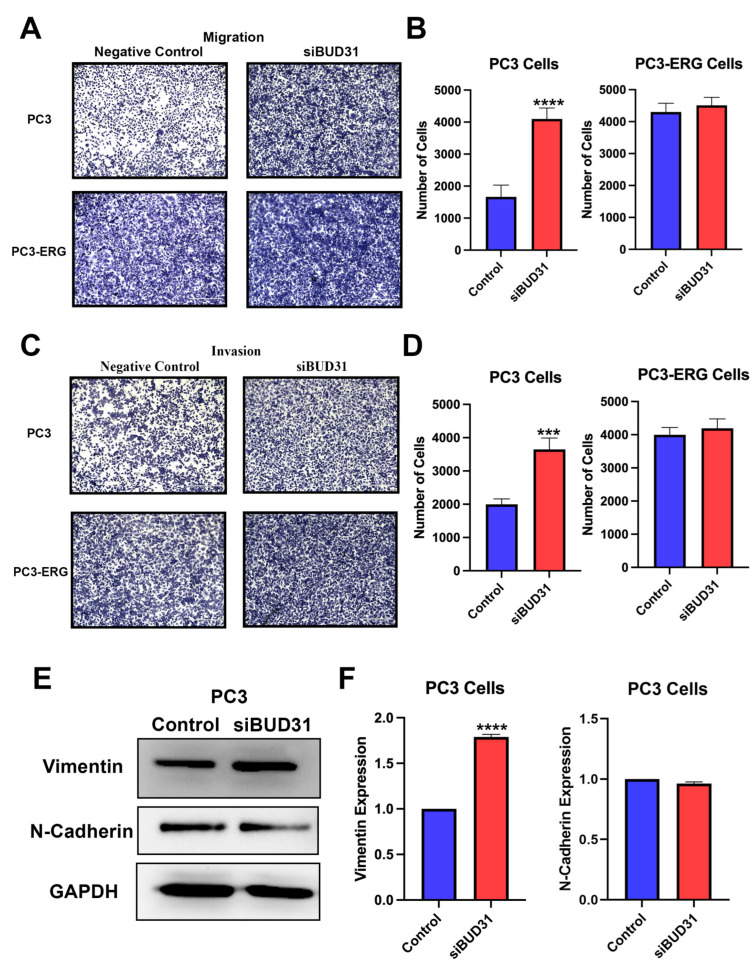
Inhibition of BUD31 enhances the ability of prostate cancer cells to migrate and invade. (**A**) Migration assay results for PC3 and PC3-ERG cell lines treated with siBUD31 and scramble RNA (negative control) viewed under 4× magnification and scale bar denotes 400 µm, performed from 3 replicates. (**B**) Quantification of migratory cell population from the migration assay for PC3 and PC3-ERG cells. (**C**) Imaging of invasion assay results for PC3 and PC3-ERG cell lines in both control and BUD31 knockdown treatment groups from 3 replicate groups. (**D**) Graphical representation of invasive cell counts in control and siBUD31-treated PC3 and PC3-ERG cells. (**E**) Western blot analysis for expression of vimentin and N-Cadherin EMT markers associated with migration and invasion. (**F**) Bar graph for quantitative demonstration of differential expression for EMT-related genes analyzed by Western blot in Figure 4E (*p* < 0.0001). (***) is *p*-value ≤ 0.001 and (****) is *p*-value ≤ 0.0001.

## Data Availability

Clinical RNAseq data in this study is extracted from the prostate adenocarcinoma (PRAD) database from The Cancer Genome Atlas (TCGA) program available to the general population.

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
