# Peer review of "Downregulation of BUD31 Promotes Prostate Cancer Cell Proliferation and Migration via Activation of p-AKT and Vimentin In Vitro"

_ijms, 2023, doi:10.3390/ijms24076055_

Round 1
Reviewer 1 Report
The authors of this study entitled "Downregulation of BUD31 promotes prostate cancer cell proliferation and migration via activation of p-AKT and Vimentin in vitro” are trying to explore the regulation of BUD31 in the proliferation of PCs cancer cells in-vitro and clinical samples. This is a very helpful study for researchers to explore new avenues for the management of prostate cancers in humans. This study could be suitable for publication with significant major revision with the following comments.
Comments :
- The resolution of the following figures 1, 2, 3, and 4B, is very poor since unable to draw any valid conclusions as hypothesized. Please provide images of 300 dpi or more.
- It would be helpful if the authors summarises the study outcome through one schematic diagram/graphical abstract in the manuscript.
- It would be recommended to provide brief information on the impact of PCs on reproductive function in the introduction/discussion part of the Manuscript.
- Line 80: Please provide TMAs abbreviation (however you gave it in the MM section).
- Line 87: Please provide the project ethics information.
- Line 95: Please provide the reference of this protocol.
- Line 102: Please provide the manufacturer’s details.
- Please provide the washing step details after antibodies incubation/provide the reference.
- Figure 1: Please provide the scale bars in all the images.
- Figure 1: There is no statistical significance information in the expression level figure panel.
- Figure 4D: Please align the quantitative bar diagram according to the western result.
- Figure 5E: It seems that the cadherin expression is decreased whereas quantitatively no change. Please explain.
- Please abbreviate ERG cell lines
Author Response
Reviewer 1
- The resolution of the following figures 1, 2, 3, and 4B, is very poor since unable to draw any valid conclusions as hypothesized. Please provide images of 300 dpi or more.
Thank you for notice. Figures have been updated to 300 dpi, this may not be prevalent in word document so you will have to refer to .tif file provided for each figure.
- It would be helpful if the authors summarises the study outcome through one schematic diagram/graphical abstract in the manuscript.
Graphical Abstract has been added.
- It would be recommended to provide brief information on the impact of PCs on reproductive function in the introduction/discussion part of the Manuscript.
Thank you. We added into the introduction detailing the effect of PCa on reproductive function. However, the impact of prostate cancer on reproductive function was not added to discussion since it was not a focus of the study and was not investigated.
“The prostate gland, a component of the male reproductive system, is where prostate cancer develops. Prostate cancer can have a variety of effects on reproductive function, according to scientific research. Studies have found infertility can result from prostate cancer therapies such surgery, radiation therapy, and hormone therapy because they can have a negative impact on sperm production and ejaculation [8]. PCa itself may also restrict the seminal vesicles, resulting in a reduction in the quantity of semen and sperm cells [9]. Finally, prostate cancer may also interfere with erectile function, making it more difficult to get and keep an erection, which may also result in infertility. Generally, the ability of men to reproduce and have children can be significantly impacted by prostate cancer [10].”
- Line 80: Please provide TMAs abbreviation (however you gave it in the MM section).
Added to abbreviation list and in the introduction.
- Line 87: Please provide the project ethics information.
Provided at the bottom in the Institutional review board statement section (page 13).
- Line 95: Please provide the reference of this protocol.
Added “DAKO Omnis Stainer User Manual. (2019). Agilent Technologies.”
- Line 102: Please provide the manufacturer’s details.
Manufacturer details have been added.
- Please provide the washing step details after antibodies incubation/provide the reference.
Washing steps have been added as per reviewers’ suggestion.
- Figure 1: Please provide the scale bars in all the images.
Scale bar added as per reviewers recommendation.
- Figure 1: There is no statistical significance information in the expression level figure panel.
Statistical significance has now been denoted in the figure using stars.
- Figure 4D: Please align the quantitative bar diagram according to the western result.
Alignment has now been corrected.
- Figure 5E: It seems that the cadherin expression is decreased whereas quantitatively no change. Please explain.
The E-Cadherin band for siBUD31 treatment is initially does seem down regulated however densitometry reveals no statistically significant change, this may be due to larger band size of siBUD31 band as it is longer therefore the expression could be even due to this size difference.
- Please abbreviate ERG cell lines
ERG is not a independent cell line. But PC3 cells overexpressed with ERG gene. Authors would like to use PC3-ERG as cell line name for ease of use and understanding for reading.

Reviewer 2 Report
Summary:
BUD31 is involved in the pre-mRNA splicing process as it is a spliceosomal component required for spliceosome assembly and catalytic activity. Studies have indicated that BUD31 may play a role as a positive regulator of AR transcriptional activity. However, the association of BUD31 expression with prostate cancer progression is still unknown. In this manuscript, Muhammad Choudhry, et al investigated the clinical implications of varying outcomes relative to BUD31 expression levels by using a cohort of TMAs and data from The Cancer Genome Atlas (PRAD-TCGA). Then they further elucidate the function of BUD31 via loss-of-function assay in prostate cancer cell lines. Data suggested that BUD31 may act as a tumor suppressor by inhibiting proliferation and migration of prostate cancer cells.
General comments:
1. The introduction does not provide sufficient background with respect to the development and progression of prostate cancer.
2. The resolution of all images is extremely low, hindering the understanding of readers to the data.
3. The number of biological replicates and significant differences between experimental groups are not indicated in all bar plots.
Author Response
Reviewer 2
- The introduction does not provide sufficient background with respect to the development and progression of prostate cancer.
The following has been updated in the introduction.
“PCa is a common malignancy in men, and its development and progression are complex processes that involve multiple genetic and environmental factors. Research has shown that the development of prostate cancer is associated with changes in the androgen signaling pathway [3], alterations in DNA repair mechanism [4], and dysregulation of various cellular signaling pathways [5]. The progression of prostate cancer from an androgen-dependent state to an androgen-independent state is a critical step in the development of advanced disease, and recent studies have shown that this transition is mediated by the activation of various growth factor signaling pathways, including the PI3K/Akt/mTOR pathway [3]. Additionally, research has identified several genetic and epigenetic alterations that contribute to the development and progression of prostate cancer, including alterations in the tumor suppressor genes PTEN and TP53 [6], and the oncogene MYC [7].”
- The resolution of all images is extremely low, hindering the understanding of readers to the data.
Figures have been updated to 300 dpi, this may not be prevalent in word document so you will have to refer to .tif file provided for each figure.
- The number of biological replicates and significant differences between experimental groups are not indicated in all bar plots.
The number of biological replicates has now been added in the figure captions of all figures.
Round 2
Reviewer 1 Report
Thank you for addressing all the suggested comments. The resolution of figure 3 is still not very good. However, I am unable to see the individual tiff files. if so, Please improve it before publication, after this revision manuscript is suitable for publication.
Reviewer 2 Report
The manuscript has been sufficiently improved to warrant publication in IJMS.